# Computer-Aided Image Enhanced Endoscopy Automated System to Boost Polyp and Adenoma Detection Accuracy

**DOI:** 10.3390/diagnostics12040968

**Published:** 2022-04-12

**Authors:** Chia-Pei Tang, Chen-Hung Hsieh, Tu-Liang Lin

**Affiliations:** 1Division of Gastroenterology, Department of Internal Medicine, Dalin Tzu Chi Hospital, Buddhist Tzu Chi Medical Foundation, Chiayi City 62224, Taiwan; franktg@hotmail.com; 2School of Medicine, Tzu Chi University, Hualien City 97004, Taiwan; 3Department of Management Information System, National Chiayi University, Chiayi City 600023, Taiwan; a0903355322@gmail.com

**Keywords:** colonoscopy, narrow-band image, colon polyp, Retinex, gamma and sigmoid conversion, YOLO

## Abstract

Colonoscopy is the gold standard to detect colon polyps prematurely. Early detection, characterization and resection of polyps decrease colon cancer incidence. Colon polyp missing rate remains high despite novel methods development. Narrowed-band imaging (NBI) is one of the image enhance techniques used to boost polyp detection and characterization, which uses special filters to enhance the contrast of the mucosa surface and vascular pattern of the polyp. However, the single-button-activated system is not convenient for a full-time colonoscopy operation. We selected three methods to simulate the NBI system: Color Transfer with Mean Shift (CTMS), Multi-scale Retinex with Color Restoration (MSRCR), and Gamma and Sigmoid Conversions (GSC). The results show that the classification accuracy using the original images is the lowest. All color transfer methods outperform the original images approach. Our results verified that the color transfer has a positive impact on the polyp identification and classification task. Combined analysis results of the mAP and the accuracy show an excellent performance of the MSRCR method.

## 1. Introduction

Colonoscopy is considered as the standard method for the diagnosis and surveillance of colon polyps. Subsequent polypectomy after colonoscopy is the most effective colorectal cancer (CRC) prevention [1]. Early colonoscopy detection and removal of polyps reduces the incidence of colorectal cancer (CRC) by 76% [2]. The most common polyps are hyperplastic and adenomatous. According to the American Society for Gastrointestinal Endoscopy, the “resect and discard” and “diagnose and leave” strategies propose that the hyperplastic polyp need not to be removed. Since hyperplastic polyps are most diminutive and non-malignant, these strategies save a great deal of resection time and pathologic analysis cost [3,4]. The traditional white light (WL) colonoscopy yields an adenoma miss rate of 26%, especially for those <5 mm in size [5,6]. Adenomatous polyps are the primary lesion which evolve to CRC and develop to an interval cancer missed in an initial colonoscopy. The identification and resection of adenomatous polyp is essential to prevent CRC [7]. Innovative methods have been introduced to decrease polyp and adenoma miss rate [8,9].

As the field of colonoscopy technology thrives, new diagnostic modalities have been introduced to improve polyp detection. Image-enhanced endoscopy (IEE) is one of the state-of-the-art tools. Digital IEE includes Olympus narrowed-band imaging (NBI), PENTAX i-scan, and FUJI linked-color imaging (LCI), which improves the diagnostic ability by enhancing polyp mucosa microstructure and microvasculature. The Olympus NBI filters the specific wavelengths to enhance mucosa and vascular pattern. PENTAX i-scan is a real time post-image software-driven modification of contrast, hue and sharpness to enhance polyp mucosa. The NBI remains the most adopted and widely used method at present [10].

NBI incorporated Olympus colonoscopy has a superb ability to detect and identify hyperplastic and adenomatous polyps. This one-button-activated electrical system is an innovative image technology and aids endoscopists to better detect and characterize polyps [11]. NBI technology allows only blue and green lights to pass through a filter placed in the colonoscope light source. The NBI wavelength of the trichromatic optical filters is between 415 and 540 nm with a bandwidth of 30 nm, which has a shallow penetration depth [12]. Two peaks of tissue hemoglobin are absorbed with the wavelength at 415 nm (blue light) and 540 nm (green light) [13]. The narrowed spectrum light highlights the mucosa surface microvasculature pattern to differentiate non-neoplastic (hyperplastic) from neoplastic (adenoma) polyp [14,15] (Figure 1). However, the bowel content of fecal material, debris or filthy water appears bright red color in contrast to the deep dark brown normal mucosa in the NBI environment, which is a visual irritant. The full-time activated NBI system might trigger visual fatigue and discomfort owing to the high color contrast image. Studies indicate that NBI increases polyp and adenoma detection rate with the full-time activated system [12,16,17,18]. In the real world, endoscopists only activate the NBI system in the circumstances of analyzing the type and margin of the polyp. Switching between WL and NBI back and forth during the withdrawal phase in a colonoscopy is time consuming and not cost-effective. As a consequence, endoscopists leave the system off in most of the colonoscopy observation period. The polyp and adenoma detection rates are not increased with the NBI system in daily clinical practice. In the era of artificial intelligence, we can overcome this issue with a tailor-made image enhancement CNN model to boost the polyp detection and classification without affecting an endoscopist’s routine performance. 

The output connection of the Olympus NBI system from the colonoscopy equipment to an external computer is not feasible. We need to convert the original WL image from the colonoscopy source to an NBI simulated CNN-based model on the background and show the WL image with the bounding box in the monitor (Figure 2). We selected three methods to simulate the NBI system for image enhancement: Color Transfer with Mean Shift (CTMS), Multi-scale Retinex (MSR), and gamma and sigmoid conversions. We also compared the selected methods with two conventional image enhancement methods, Histogram Equalization (HE) [19,20] and Contrast Limited Adaptive Histogram Equalization (CLAHE) [21].

The CTMS conversion process is inspired by the work of Xiao et al. [22], who transferred the insufficient training dataset images from the RGB color space to the CIE-LAB color space using a U-Net architecture, to generate the data augmented images. 

MSR algorithm is an image enhancement method mimicking human visual perception which provides dynamic range compression, tonal rendition and color constancy [23,24,25,26]. Our eyes perceive colors by the light reflection back from an object with a certain wavelength. The human visual system captures colors irrespective of the illumination source under different spectral lighting conditions from a scene. The MSR algorithm separates the original image into a base and a detailed layer, which are processed to improve nonuniform illumination [27]. It has been used for various issues as image dehazing [28], image enhancement and defogging [29] and color constancy computation [30]. In real-world colonoscopy images, the illumination varies with uneven darkness and brightness owing to the light source on the tip of the colonoscope [31]. Luo et al. [32] used a modified MSR with detailed layer to solve the nonuniform and directional illumination on the surgical endoscopy field. Their combined visibility was improved from 0.81 to 1.06 and outperformed existent Retinex methods. Wang et al. [33] corrected color images based on a MSR with a nonlinear functional transformation. They improved the overall brightness and contrast of an image and preserved the image details. Vani et al. [31] discussed the use of MSR and Adaptive Histogram Equalization to suppress noise and improve visibility in wireless capsule endoscopy. Deeba et al. [34] proposed a two-stage automated algorithm with Retinex and saliency region detection algorithm. They achieved a sensitivity of 97.33% and specificity of 79%. MSR provides superb endoscopy image enhancement with balanced brightness and contrast to detect subtle lesions in colonoscopy.

The sigmoidal-remapping function is accomplished by enhancing the image contrast in the limited dynamic range. That is, the lightness between the highlight and shadow in an image can be controlled with the lightness and darkness of the contrast in the sigmoid function [35]. The sigmoidal-remapping function is a continuous nonlinear activation curve [36]. Deeba et al. [37] used a sigmoidal remapping curve to enhance the blue and green light channels in the endoscopy image combined with saliency map formation and histogram of gradient for feature extraction. They achieved a recall and F2 score of 86.33% and 75.51%, respectively.

In this study, we aim to establish a NBI simulated image enhancement technique combined with the computer-aided system to boost polyp detection and classification. We chose three different methods and compared them to each other for their effectiveness in endoscopy image enhancement.

## 2. Materials and Methods

### 2.1. Materials

The colonoscopy images were taken from colonoscopies performed with high-definition colonoscopes (CF-H290I, Olympus, Tokyo, Japan) in Dalin Tzu Chi hospital, a teaching hospital in Taiwan, from December 2021 to March 2022, with the approval of the Institutional Review Board (B11004010).

#### 2.1.1. Dataset

The polyp dataset for the training of the deep learning network model was divided into two parts according to the obtaining method. The first part was the static colonoscopy images that were manually selected and captured. Most of the manually selected polyp images were clear compared with those captured from colonoscopy videos. The manual selection process ensured the better image quality and avoided the similarity of polyp images. There were a total of 3796 images, of which 3693 images had more than one polyp. The remaining 103 images were background images which did not contain polyp. The second part of the dataset was extracted from 25 recorded colonoscopy videos performed by senior endoscopists. The total duration of the 25 videos was about 3.1 h (around 336,780 frames), and the number of detected polyps in each video varied. There were 3 complete colonoscopy inspection videos, and the remaining 22 were the segments of detected polyp videos. After deleting the unrecognizable images, 2719 images were included in this study and 1347 images were without polyps. The images were stored with a resolution of 1920 × 1080. 

The dataset was divided into three categories according to the types of polyp, i.e., hyperplastic polyp (HP), tubular adenomatous polyp (TA), and sessile serrated adenoma (SSA). The TA and SSA polyps require resection and the HP polyps are considered to not need resection during colonoscopy. There were 1486 images of HP, 2687 images of TA, and 892 images of SSA polyps. Table 1 shows the statistics of the images.

#### 2.1.2. Data Labeling

Polyps were labeled using the LabelImg image label tool in this study. The labeled images in this study were divided into three categories. The first step was to label all polyps in the dataset, which was to identify the presence of polyp in the image. The identified polyps were divided into three types: TA, HP, and SSA.

#### 2.1.3. Data Augmentation

Data augmentation is a common technique in object recognition. By scaling, cropping, and rotating, the amount of training data for the model training increases to improve the accuracy of the model. In Yolo v4 network training, the image is randomly rotated by plus or minus 180 degrees. The hue and saturation are adjusted. The images are randomly scaled, cropped, and collaged with Mosaic’s data augmentation method for training.

### 2.2. Methods

#### 2.2.1. Color Transfer with Mean Shift

Xiao et al. proposed a novel Color Transfer with Mean Shift method, a data augmentation technique to improve the performance of the deep learning network model for small data. Xiao et al. transferred the training data from the RGB color space to the CIE-LAB color space, a color space defined by the International Commission on Illumination (CIE), through a matrix. The method proposed by Xiao et al. selected a target image and calculated the color mean value of the target image; the mean value is imposed on the original image to generate a new image. The process is formulated as Equation (1). C_transferred_ represents the converted value, C_original_ represents the value of the image to be converted, ${\bar{C}}_{\mathrm{original}}$ represents the mean value of channel C calculated from the main coloring area of the original image, and ${\bar{C}}_{\mathrm{target}}$ represents the mean value of channel C in the transferred area of the target image.
(1)$$C_{\mathrm{transferred}}=C_{\mathrm{original}}\;-\;{\bar{C}}_{\mathrm{original}}\;+\;{\bar{C}}_{\mathrm{target}}$$

Their study proved that the proposed data augmentation method had better performance than the traditional geometric data augmentation methods (scaling and rotation). The network model trained with this data augmentation method generalized better [22].

We applied the Color Transfer with Mean Shift to the polyp detection. For mimicking the features of NBI colonoscopy image using WL images, we converted the channels A* and B* using Equation (2). The C is the converted channel value, C_WL_ represents the channel value of WL colonoscopy images, ${\bar{C}}_{\mathrm{WL}}$ represents the mean value of WL colonoscopy image channels, ${\bar{C}}_{\mathrm{NBI}}$ represents the mean value of channels in NBI colonoscopy image, x is the coefficient to adjust the NBI image value of the channel, and y is the constant to fine-tune the color tone.
(2)$$C_{\mathrm{transferred}}=C_{\mathrm{WL}}\;-\;{\bar{C}}_{\mathrm{WL}}+x\;*{\bar{C}}_{\mathrm{NBI}}+y$$

Figure 3 and Figure 4 are the color transfer examples of two types of polyps, adenomatous and hyperplastic polyps. Figure 3 and Figure 4 show the images of TA and HP polyps after Color Transfer with Mean Shift.

#### 2.2.2. Retinex

Retinex is a common method for image enhancement, a term combining retina and cortex. Retinex minimizes the effect of a light source on the image to achieve color constancy. Retinex has three steps to maintain the color constancy. First, the image is dynamically compressed to preserve the details of the original image. The second step is to isolate the color from the spectrum of the scene’s light source. Finally, the color of the object in the image is restored and reproduced. 

We formulate the above mentioned three steps into the following mathematic expressions. Each pixel in the image is expressed as the product of the light source intensity and the reflection intensity as Equation (3), where *S* is the pixel value of coordinate (x,y) in the image, *R* is the reflection expression, and L is the intensity of the light source.
(3)$$S\left(x,y\right)=R\left(x,y\right)*L\left(x,y\right)$$

Dynamic range compression is used to compress the original signal into a smaller range. In the digital image, the signal is compressed into the range of the according signal. To obtain the details in the image, Retinex dynamically compresses the image logarithmically as Equation (4).
(4)$$\log\left(R\left(x,y\right)\right)=\log\left(S\left(x,y\right)\right)-\log(L\left(x,y\right)$$

The natural color is independent of the spectrum of the light source. The goal of Retinex is to eliminate the influence of a different light source intensity. The elimination is expressed as Equation (5), where *R_i_* is the result of the output on channel *I, I_i_* is the pixel value of channel *i* in the image. *F* is the Gaussian Surround Function that simulates the illumination of light sources in nature, as Equation (6); c is the Gaussian surround space constant.
(5)$$R_{i}\left(x,y\right)=\log\left[I_{i}\left(x,y\right)\right]-\log\left[F\left(x,y\right)*I_{i}\left(x,y\right)\right]$$
(6)$$F\left(x,y\right)=e^{-r^{2}/c^{2}}$$

When the Gaussian surrounding space constant is small (*c* < 20), it has an improved dynamic range compression effect and retains more image details. When the constant is increased (*c* > 200), better color restoration is achieved [38]. The Gaussian surrounding space constant is used only once in the Single Scale Retinex (SSR); trade-off is made between the two. Therefore, Multi-Scale Retinex (MSR) was developed, which uses multiple Gaussian surrounding space constants in the image to gain the advantages of different scales simultaneously, as in Equation (7).
(7)$$R_{MSR_{i}}=\sum_{n=1}^{N}w_{n}R_{ni}$$
(8)$$F_{n}\left(x,y\right)=ke^{-r^{2}/c_{n}^{2}}$$

$R_{MSR_{i}}$ in Equation (7) is the output of the *i*th channel of MSR, $w_{n}$ is the weight, $R_{ni}$ is the result of SSR output using $c_{n}$ as the Gaussian surrounding space constant, and Equation (8) is the Gaussian surrounding function in MSR. In several experiments, it is shown that three scales are sufficient for MSR, a small scale ($c_{n}$ < 20), a large scale ($c_{n}$ > 200), and an intermediate scale. The weights were assigned equally for the 3 scales, i.e., 1/3 of each scale [23].

Originally, the Retinex algorithm was based on the Gray-World Assumption. As the reflectance of the image is the same in all three primary color channels, it meets the Gray-World Assumption. This assumption is violated when the image is not colorful or has a large number of single blocks of color, which results in Retinex being grayed out or having severely reduced saturation. Therefore, D. J. Jobson proposed that adding a color restoration function to the MSR and converting it to the Multi-Scale Retinex with Color Restoration (MSRCR) as in Equations (9) and (10), where *β* = 46, *α* = 125, *b* = −30, and *G* = 192 [38].
(9)$$R_{MSRCR_{i}}\left(x,y\right)=G\left[C_{i}\left(x,y\right)R_{MSR_{i}}\left(x,y\right)+b\right]$$
(10)$$C_{i}\left(x,y\right)=\beta \log\left[\alpha I_{i}^{\prime }\left(x,y\right)\right]=\beta \log\left[\alpha I_{i}\left(x,y\right)\right]-\beta \log[\sum_{i=1}^{S}I_{i}\left(x,y\right)]$$

Parthasarathy and Sankaran improved MSRCR by proposing an automated multiscale Retinex for color restoration, adding a hue conversion function to the MSRCR output. They used a histogram to calculate color pixel thresholds, limiting all to two thresholds [39].

Figure 5, Figure 6 and Figure 7 are the Retinex examples of three polyp types, TA, HP, and SSA. 

#### 2.2.3. Gamma and Sigmoid Conversions

Deeba et al. proposed a computer-aided polyp detection algorithm for wireless capsule endoscopy [37]. Their proposed method is divided into three steps. The image is enhanced, followed by the generation of salient graphics to highlight the location of possible polyps, and finally the histogram of gradient values (HOG) is calculated for feature extraction. Deeba et al. enhanced the blue and green light channels by using the sigmoid curve in the image, as in Equation (11).
(11)$$I_{s}=I_{MAX}*\frac{1}{1+e^{-a\left(I-c\right)}}$$

The values at both ends of the sigmoid curve are compressed, and the values in the middle are stretched between 0 and 1. The $I_{s}$ is the result of sigmoid calculation, $I_{MAX}$ is the maximum pixel value in a single channel, *a* and *c* are constants, the size of the constant is adjusted according to the image, and the red channel is suppressed using the Transform-based Gamma Correction (TGC) curve in Equation (12).
(12)$$I_{G}=\left(I_{MAX}-S\right)*{\left(I/I_{MAX}\right)}^{r}$$

The $I_{G}$ is the calculated result of the gamma curve and S is a constant that controls the effect of curve suppression [37]. Figure 8 shows the gamma and sigmoid conversion of RGB channels. Figure 9 shows the images after adjusting the parameters of the sigmoid curve and gamma correction curve using colonoscopy images.

#### 2.2.4. YOLOv4

YOLOv4 is a one-stage object recognition method proposed by Alexey Bochkovskiy et al. in 2020 [40]. Based on Yolov3 [41], YOLOv4 has made improvements in various areas. YOLOv4 has improved both the speed and accuracy of recognition compared with previous versions. The recognition AP of MS COCO dataset on Tesla V100 GPU reaches 43.5%, and the recognition speed is about 65FPS. The network model of object recognition is divided into four parts according to their different functions, which are the input layer responsible for image input, the backbone of the main body of the object recognition network, the neck connecting the backbone and the head, and the layer responsible for classification and bounding boxes. The head of YOLOv4 follows the original YOLOv3, the trunk uses CSPDarknet53 previously developed by the author, and the neck uses Spatial Pyramid Pooling (SPP) and Path Aggregation Network (PANet).

YOLOv4 refers to several of the latest object recognition methods mentioned on the Browse State-of-the-Art website in the field of object recognition, which are applied to various parts of the network to evaluate their quality. YOLOv4 selects the best performance method and was further improved. The author conducted experiments and improvements on six parts, including data amplification, activation function, bounding box regression loss, normalization, normalization of network activation through mean and variance, and skip-connections. After experiments and comparisons, YOLOv4 finally added CutMix, Mosaic data amplification, Class Label Smoothing, DropBlock, Mish activation function, Cross-stage partial connections and Multi-input weighted residual connections to the backbone part; and CIoU loss, CmBN, DropBlock, Mosaic data augmentation, Self-Adversarial Training, Eliminate grid sensitivity, Cosine annealing scheduler, Optimal hyperparameters, Mish activation function, SPP, SAM, PANet, and DIoU to the head section—technologies such as NMS.

In addition to the improvement in accuracy and speed, YOLOv4 proves that complete training on a general consumer-grade graphic card is possible, making the YOLOv4 algorithm more widely adopted [40].

#### 2.2.5. Research Design

In this research, the colorectal polyp identification is performed using two experimental designs. In the first experimental design (Figure 10), the color transfer is performed on the entire colonoscopy image. In the implementation, we found that when the color transfer is performed on the entire high-resolution image, it takes some extra time to convert every pixel of the entire image. However, in the actual diagnosis process, it is expected that the system can generate instant results. Therefore, in order to improve this immediacy problem, this study further proposes a second experimental design (Figure 11). First, the WL endoscopic image is used to identify the polyp and performs color transfer in the identified polyp areas. The entire process takes less time and enhances its immediacy due to a smaller area for color transfer.

Figure 10 is the structure diagram of the first experimental design. The first step is the collection and arrangement of the dataset. The video of the colonoscopy is captured into static images. Poor quality images are filtered out. They are merged with the static images of the colonoscopy. The LabelImg tool is adopted to label the location and type of polyps. The second step is pre-processing, converting the labeled data using the color transfer function, then randomly splitting the data into a training dataset and a testing dataset in a ratio of 8 to 2, and using K-means to convert the object frames into groups. The third step is to add the data into the deep learning network for training and evaluate the network model with the test dataset. 

The second experiment design is to add an image classification network of Efficientnet v2-m after obtaining the polyp identification results from the first experiment as shown in Figure 11. In the first experiment design, we collected 11,957 images with polyps and 9369 images without polyps and trained a polyp recognition model using Yolov4 to detect polyps in images. The final mAP of the model was 92.4%. After the recognition results of Yolov4, we performed color transfer on the object frame output by the network and trained the types of polyps using the Efficientnet v2-m network. Because the color transfer is only performed on the object frames rather than the entire image, we reduced the time complexity, making polyp image recognition quicker than the first experiment.

This study attempts to remove unimportant information in colonoscopy images using a special color transfer processing method to preserve or highlight the details of polyps. Three different color transfer methods are used, Color Transfer with Mean Shift, automatic multi-scale Retinex for color restoration, and gamma and sigmoid conversion. In this study, it is believed that Color Transfer with Mean Shift and NBI have similar concepts in reducing original data, and the gamma and sigmoid conversion simulate the same pattern as NBI suppresses red light. Since the colonoscopy is illuminated by a direct single light source, the bright area is interspersed with uneven dark shadows owing to folds or fecal debris. In this study, the color and detail contours which disappeared in the image are restored using the automatic MSR algorithm.

#### 2.2.6. Model Training

This research uses YOLOv4 to train the model. First, the dataset is grouped into 9 scales that conform to the ground truth of the dataset using k-means. The model uses the calculated 9 scales as the anchor boxes size in the final output layer of the network. In the model training, the loss graph of the model is monitored until the loss becomes flat, and the training stops, that is, the model has converged.

#### 2.2.7. Evaluation Metrics

This research adopts the mean Average Precision (mAP) to measure the accuracy of the model. The calculation of mAP is an AP calculation for all categories which takes the average. mAP is used as a metric for object detection. Intersection over union (IoU) is also added to measure the correctness of the target position marked by the model. The IoU is the intersection of the model-predicted box and the ground truth divided by the union of the two boxes as in Equation (13). The calculation of AP is as in Equation (14). In short, the AP is the average precision of one category and mAP is the mean average precision of all categories. The higher the mAP value, the higher the accuracy of the search results.
(13)$$\mathrm{IoU}\left(A,B\right)=|\frac{A\cap B}{A\cup B}|$$
(14)$$\mathrm{AP}=\frac{1}{11}\sum_{r\in \left\{0,0.1\ldots ,1\right\}f}P_{interp}\left(r\right)$$

In the calculation of mAP, an IoU is usually set as the critical value (usually set to 0.5). When the IoU is greater than the preset critical value, it is classified as True Positive (TP); otherwise, it is classified as a false positive (FP). The real object is not predicted by the model classified as false negatives (FN), and the false object is not predicted as true negatives (TN). AP is calculated according to these values and estimates the area under the PR curve (Precision-Recall) (AUC). In machine learning, there are two methods to measure the model: Precision as Equation (14) and Recall as Equation (15).
(15)$$\mathrm{Precision}=\frac{\mathrm{TP}}{\mathrm{TP}+\mathrm{FP}}$$
(16)$$\mathrm{Recall}=\frac{\mathrm{TP}}{\mathrm{TP}+\mathrm{FN}}$$

In the PASCAL VOC challenge, Equation (14) is used to calculate the AP average of 11 Recall interpolation points as the AP of the object. Finally, the AP of all objects in the model is averaged to gain the mAP. The mAP in this research is calculated using the program on Yolo v4-AlexeyAB GitHub.

In the Efficientnet v2-m classification network, the accuracy is used as the evaluation standard. If the quasi-class predicted by the model is the same as the real situation, it belongs to TP or TN, and if the predicted result does not match the real situation, it belongs to FP or FN. Finally, the accuracy is calculated using the four values of the confusion matrix. As in Equation (17), the accuracy is the percentage of correct predicted results in the total sample.
(17)$$\mathrm{Accuracy}=\frac{\mathrm{TP}+\mathrm{TN}}{\mathrm{TP}+\mathrm{FP}+\mathrm{FN}+\mathrm{TN}}$$

## 3. Results

In this study, different color transfer methods were trained using deep learning networks and were compared with the model trained on the original images to evaluate the effect of color transfer methods on polyp identification and classification. The data are divided into three datasets according to different classification methods. The first dataset is divided into three types of polyps (TA, HP and SSA). The second dataset divides polyps into neoplastic (TA and SSA) and non-neoplastic (HP). The third dataset excludes the SSA due to them being scarce in number. 

### 3.1. YOLOv4 Training Results in Experimental Design 1

In this section, the three datasets use MSRCR, gamma and sigmoid conversion, Color Transfer with Mean Shift, and original images to train the model with Yolov4 network, and we compare the model results. The neoplastic polyps, TA and SSA, are grouped into the same category for color transfer comparison. We use the MSRCR to compare with other color transfer methods and it shows the highest value at 77.6162 in mAP. The results of the conventional HE and CLAHE are also included to benchmark with the selected color transfer methods. The images are first transferred from the RGB color space to LAB color space and the HE and CLAHE are performed on the lightness L channel. There are two main operational parameters, tile size and clip limit, in the CLAHE image enhancement process. The tile size is the number of the non-overlapping tiles to which the original image is partitioned and is set to 8 × 8 in the experiments. The clip limit is the threshold that will be used to trim the histograms of the pixel distribution and is set to 2 in the experiments. The MSRCR performs best in the two groups of polyps (Table 2). Therefore, MSRCR is the most suitable color transfer method to use in two classes. 

Then, the dataset is divided into three categories according to different types of polyps for further comparison of color transfer methods. The gamma and sigmoid conversion mAP have the highest value of 72.1863 (Table 3). Therefore, gamma and sigmoid conversion are suitable for color transfer in terms of the three polyp classes. 

Since the SSA is a rare polyp with insufficient data, it affects the accuracy of the identification result. We exclude the SSA category and compared the mAP results of the two categories of HP and TA for color transfer. The mAP demonstrates that the MSRCR is the most suitable color transfer method, and its mAP value is 86.8422 (Table 4). 

Comparing the results of the three color transfer methods, the MSRCR has the best mAP result in two polyp classes. Although gamma and sigmoid conversion has a better result in three polyp classes analysis, we need to consider the low probability in the SSA group owing to scarce data. By excluding the SSA, the mAP of MSRCR color transfer is higher than using gamma and sigmoid conversion with TA and SSA combined. We speculate that the higher accuracy of gamma and sigmoid conversion on SSA results in better mAP than the MSRCR in three polyp classes. After considering the distribution of the dataset and the mAP results, this study selects MSRCR as the color transfer method. 

### 3.2. Classifier Training Results in Experimental Design 2

In this section, the MSRCR, gamma and sigmoid conversion, Color Transfer with Mean Shift and original image dataset are adopted to train the model using Efficientnet v2-m network in three datasets. The polyps are automatically cropped from images and the classification task is performed based on the cropped polyp images. The accuracy is used as the criteria for polyp classification evaluation. The classification is correct if the type of polyp identified by the model matches the ground truth.

The results in the case of HP vs. combination of TA and SSA polyps can be seen from the above table (Table 5). The result shows the best accuracy of MSRCR with 0.8643. Therefore, the MSRCR is the most accurate method for color transfer classification. 

Then, the classification was performed in three polyp classes. The gamma and sigmoid conversion show the best accuracy of 0.7517, and the accuracy of MSRCR is 0.7491 (Table 6). The values are similar. In this classification analysis, both methods are suitable for the task.

For the results after excluding SSA polyps, the MSRCR accuracy is the highest with a value of 0.8428 (Table 7). Under this classification task, the most suitable color transfer method is MSRCR. 

From the accuracy comparison of the different methods in several classes, the classification accuracy of the original image is the lowest. All color transfer methods using datasets other than the original image perform better. The results demonstrated that the color transfer has a positive impact on the polyp identification. Combining the mAP results from the previous section and the accuracy results in this section, we concluded that the Retinex has an excellent outcome and is the best choice for color transfer. 

## 4. Discussion

We aim to demonstrate that color transfer is useful for the correct identification of polyps. We developed a deep learning network to predict the test dataset; three color transfer methods are compared with the base prediction of the original image dataset for the optimal identification model selection. Several image results were selected for comparison. The detected image result of MSRCR vs. Original Image are shown in Figure 12, Figure 13 and Figure 14. In the original image models of Figure 12, Figure 13 and Figure 14, the polyps were undetected or misclassified.

## 5. Conclusions

In this study, color transfer methods were successfully applied to polyp identification. The colon polyp detection accuracy and classification increased. From different experimental settings in this study, the comparison of the three color transfer methods showed that the MSRCR method outperforms others. The scarce SSA polyp images led to suboptimal results. The dataset ought to increase for the accuracy improvement of SSA detection.

We developed two research designs in this study. The first design converts each pixel of the image individually because of the nature of the color transfer. We concluded that the computation time was reduced to improve the response time. The second research design was proposed to identify the position of polyps, perform the color transfer, and classify them according to the conversion results. This ensures immediacy by reducing the area of conversion to increase the speed of the entire process.

## Figures and Tables

**Figure 1 diagnostics-12-00968-f001:**
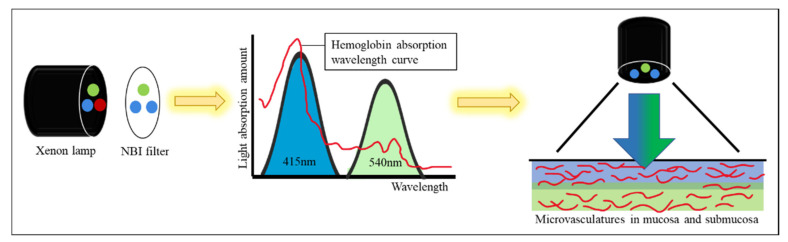
Physics of the NBI endoscopy system. The optical filter on the xenon lamp filters the red light to enhance the vascular and mucosa surface pattern.

**Figure 2 diagnostics-12-00968-f002:**
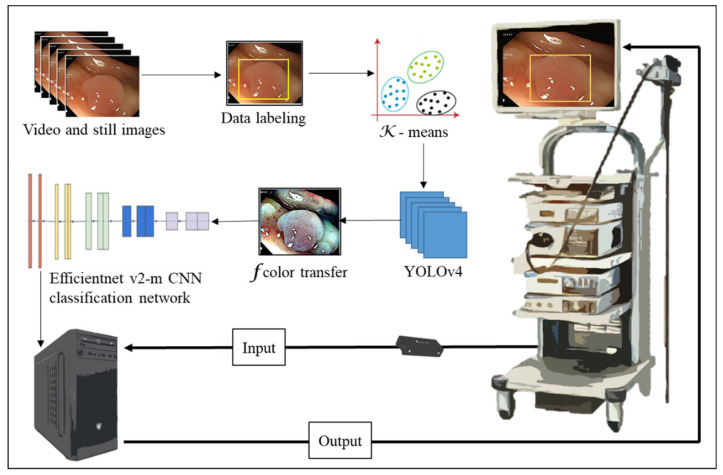
NBI simulated image enhancement system workflow. The polyp images are labeled and trained with YOLOv4 before color transfer to boost detection and classification accuracy.

**Figure 3 diagnostics-12-00968-f003:**
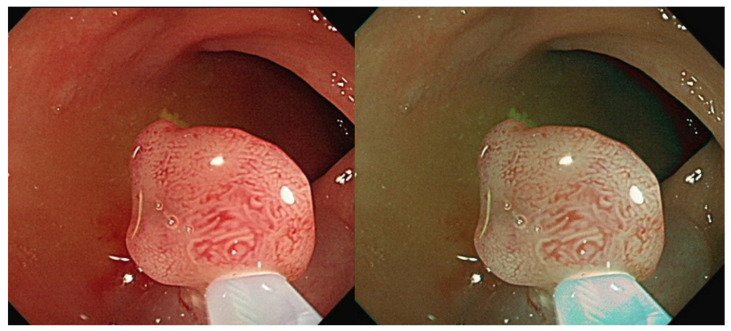
Images of TA polyps after Color Transfer with Mean Shift. ((**Left**): Original WL, (**Right**): Color Transferred).

**Figure 4 diagnostics-12-00968-f004:**
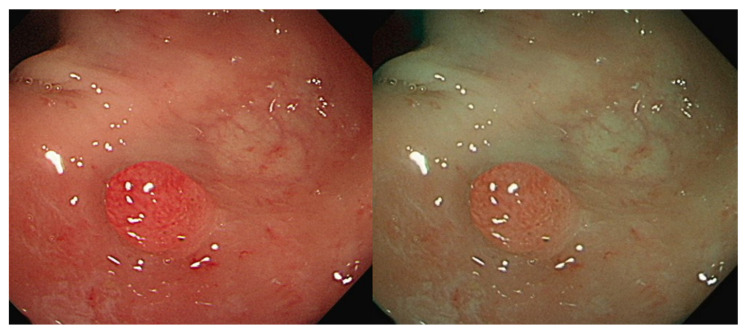
Images of HP polyps after Color Transfer with Mean Shift. ((**Left**): Original WL, (**Right**): Color Transferred).

**Figure 5 diagnostics-12-00968-f005:**
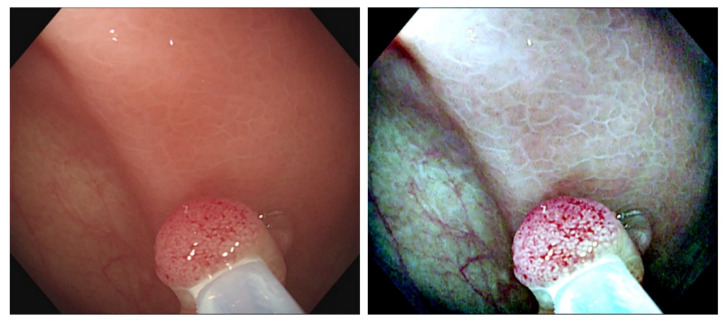
Images of TA polyp after Retinex. ((**Left**): Original WL, (**Right**): Retinex).

**Figure 6 diagnostics-12-00968-f006:**
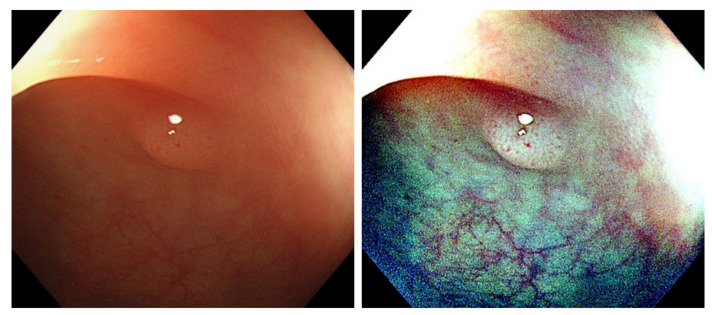
Images of HP polyp after Retinex ((**Left**): Original WL, (**Right**): Retinex).

**Figure 7 diagnostics-12-00968-f007:**
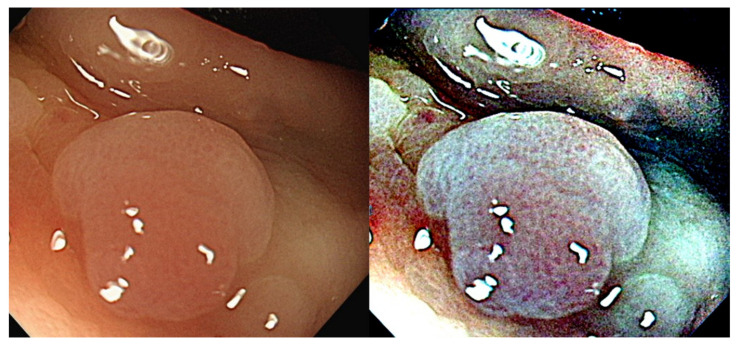
Images of SSA polyp after Retinex. ((**Left**): Original WL, (**Right**): Retinex).

**Figure 8 diagnostics-12-00968-f008:**
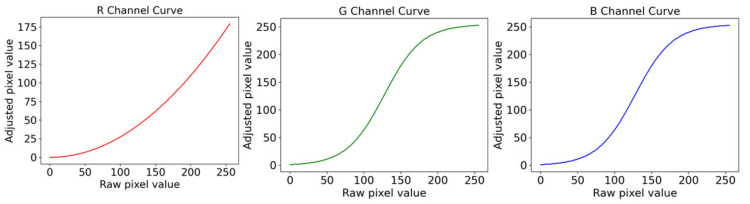
Conversion diagram of RGB in gamma and sigmoid conversion.

**Figure 9 diagnostics-12-00968-f009:**
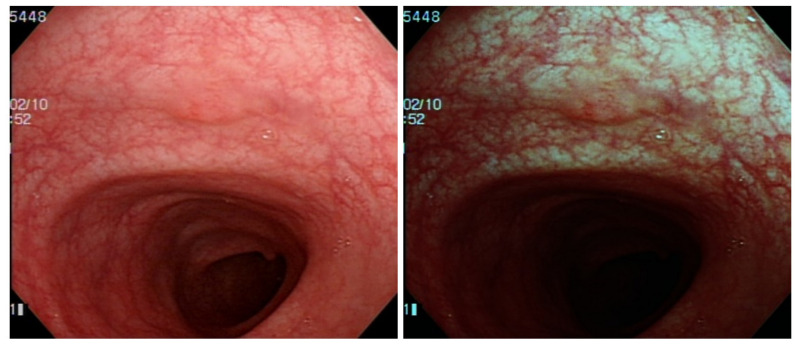
Image of SSA polyp after gamma and sigmoid conversion. ((**Left**): Original WL, (**Right**): Gamma and Sigmoid Conversion).

**Figure 10 diagnostics-12-00968-f010:**
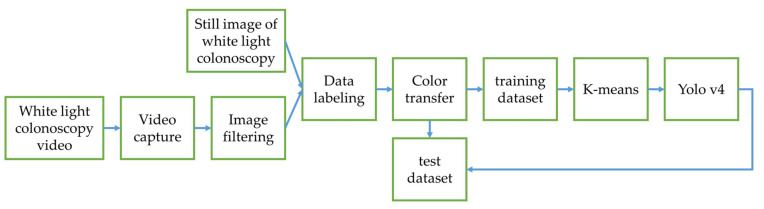
Experimental Design 1.

**Figure 11 diagnostics-12-00968-f011:**
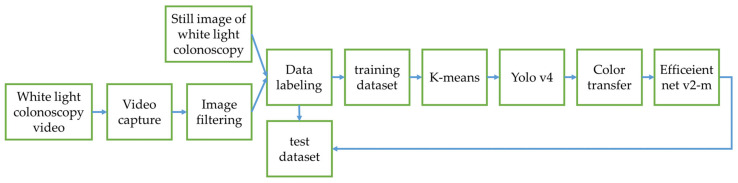
Experimental Design 2.

**Figure 12 diagnostics-12-00968-f012:**
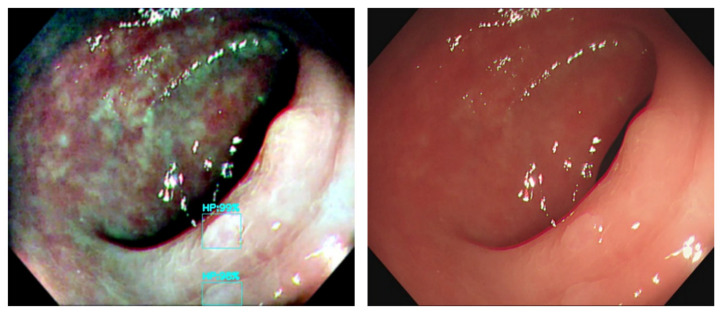
The correct results of MSRCR vs. the undetected polyp in original image. ((**Left**): MSRCR, (**Right**): Original).

**Figure 13 diagnostics-12-00968-f013:**
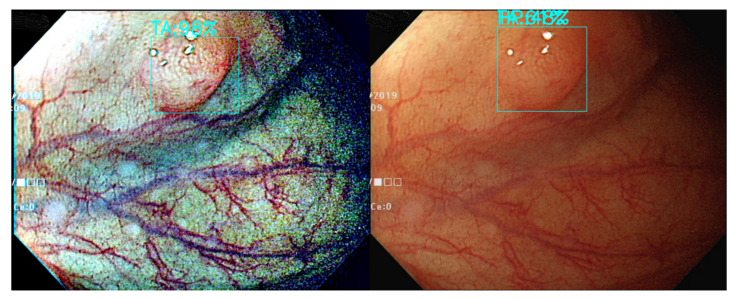
The correct results of MSRCR vs. the misclassified polyp of original image. ((**Left**): MSRCR, (**Right**): Original).

**Figure 14 diagnostics-12-00968-f014:**
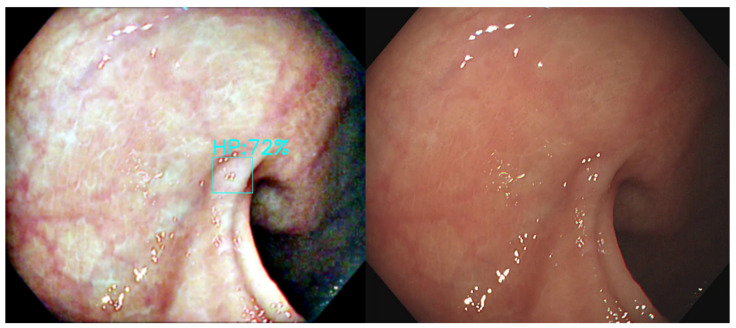
The correct results of MSRCR vs. the undetected polyp of original image. ((**Left**): MSRCR, (**Right**): Original).

**Table 1 diagnostics-12-00968-t001:** The number of images in each polyp category of the dataset.

	TA	SSA	HP	Background
Number of images	2687	892	1486	1450

**Table 2 diagnostics-12-00968-t002:** mAP results of 3 color transfer methods in HP vs. combination of TA and SSA.

Method	HP	TA+SSA	mAP
Original Image	76.18	76.08	76.1296
HE	73.86	75.89	74.8799
CLAHE	75.31	75.92	75.615
Multi-scale Retinex with Color Restoration	**78.53**	**76.70**	**77.6162**
Gamma and Sigmoid Conversions	76.54	76.38	76.4582
Color Transfer with Mean Shift	78.30	76.40	77.3478

HE: Histogram Equalization, CLAHE: Contrast Limited Adaptive Histogram Equalization.

**Table 3 diagnostics-12-00968-t003:** mAP results of 3 color transfer methods in 3 polyp classes.

Method	HP	TA	SSA	mAP
Original Image	74.80	84.09	46.17	68.3590
HE	73.42	84.4	55.24	71.0244
CLAHE	73.6	85.5	55.37	71.4944
Multi-scale Retinex with Color Restoration	75.40	86.16	48.08	69.8782
Gamma and Sigmoid Conversions	**75.69**	**87.85**	**53.02**	**72.1863**
Color Transfer with Mean Shift	75.19	84.44	45.88	68.1714

HE: Histogram Equalization, CLAHE: Contrast Limited Adaptive Histogram Equalization.

**Table 4 diagnostics-12-00968-t004:** mAP results of 3 color transfer methods in 2 polyp classes, HP and TA.

Method	HP	TA	mAP
Original Image	80.40	86.15	83.2753
HE	79.48	87.23	83.3543
CLAHE	80.56	86.24	83.3963
Multi-scale Retinex with Color Restoration	**84.43**	**89.35**	**86.8422**
Gamma and Sigmoid conversions	79.83	87.30	83.5650
Color Transfer with Mean Shift	81.71	88.07	84.8913

HE: Histogram Equalization, CLAHE: Contrast Limited Adaptive Histogram Equalization.

**Table 5 diagnostics-12-00968-t005:** Accuracy of 3 color transfer methods in HP vs. combination of TA and SSA.

Method	Accuracy
Original Image	0.8377
HE	0.8394
CLAHE	0.8377
Multi-scale Retinex with Color Restoration	**0.8643**
Gamma and Sigmoid Conversions	0.8533
Color Transfer with Mean Shift	0.8502

HE: Histogram Equalization, CLAHE: Contrast Limited Adaptive Histogram Equalization.

**Table 6 diagnostics-12-00968-t006:** Accuracy of 3 color transfer methods in 3 polyp classes.

Method	Accuracy
Original Image	0.7257
HE	0.7474
CLAHE	0.7487
Multi-scale Retinex with Color Restoration	0.7491
Gamma and Sigmoid Conversions	**0.7517**
Color Transfer with Mean Shift	0.7474

HE: Histogram Equalization, CLAHE: Contrast Limited Adaptive Histogram Equalization.

**Table 7 diagnostics-12-00968-t007:** Accuracy of the 3 color transfer methods in 2 polyp classes, HP and TA.

Method	Accuracy
Original Image	0.8017
HE	0.8200
CLAHE	0.8384
Multi-scale Retinex with Color Restoration	**0.8428**
Gamma and Sigmoid Conversions	0.8394
Color Transfer with Mean Shift	0.8308

HE: Histogram Equalization, CLAHE: Contrast Limited Adaptive Histogram Equalization.

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
