# Peer review of "Computer-Aided Image Enhanced Endoscopy Automated System to Boost Polyp and Adenoma Detection Accuracy"

_diagnostics, 2022, doi:10.3390/diagnostics12040968_

Round 1
Reviewer 1 Report
The authors tested the effect of three different methods of image enhancement to boost polyp detection and characterization in the framework of colonoscopy imaging.
They conclude that Multi-scale Retinex with Color Restoration (MSRCR) achieves better performances. Overall the work is interesting even if the obtained improvement with respect to using the original images appears modest. Moreover, I found some flaws that prevent its publication in the present form.
They are listed below:
-The authors cite in the introduction others methods and their performances. However, a comparison between the proposed methods and preexisting ones is not provided nor discussed.
- Figure captions are too short. I recommend enlarging the description of what is represented in each figure
- Figures 2 and 8 have a low resolution. The authors should improve it. Moreover, Figure 8 lacks axis labels.
- It is not clear to me what Figure 12 represents. In my opinion, the authors should either improve the figure or remove it.
- Finally, the authors should carefully proofread the manuscript before publication.
Author Response
Dear Editors,
We would like to submit the revised manuscript entitled “Computer-aided image enhanced endoscopy automated system to boost polyp and adenoma detection accuracy”, which we wish to be considered for publication in Diagnostics.
The followings are our responses to the reviewer's comments.
(1)Reviewer’s Comment: -The authors cite in the introduction others methods and their performances. However, a comparison between the proposed methods and preexisting ones is not provided nor discussed.
Authors’ Response: We thank the reviewer for the comment. We also compared the selected methods with two conventional image enhancement methods, Histogram Equalization (HE) and Contrast Limited Adaptive Histogram Equalization CLAHE. The results of the conventional HE and CLAHE are also included to benchmark with the selected color transfer methods in Table 2, 3, 4, 5, 6, 7.
(2)Reviewer’s Comment: Figure captions are too short. I recommend enlarging the description of what is represented in each figure
Authors’ Response: We thank the reviewer for the comment. We have enlarged the description of the figures as the reviewer recommended.
(3)Reviewer’s Comment: Figures 2 and 8 have a low resolution. The authors should improve it. Moreover, Figure 8 lacks axis labels.
Authors’ Response: We thank the reviewer for the comment. We have increased the resolution of the figures and add the axis labels as the reviewer recommended.
(4)Reviewer’s Comment: It is not clear to me what Figure 12 represents. In my opinion, the authors should either improve the figure or remove it.
Authors’ Response: We thank the reviewer for the comment. We have removed the Figure 12.
(5)Reviewer’s Comment:
Finally, the authors should carefully proofread the manuscript before publication.
Authors’ Response: We thank the reviewer for the comment. We have proofread the paper to make sure that the grammar errors are greatly reduced. We will send the paper to editing service to make sure the material is ready for publishing. We thank again for the reviewer’s comment, and we found the comments are very helpful to promote the quality of our paper.
We thank again for the reviewer’s comment, and we found the comments are very helpful to promote the quality of our paper.
Sincerely,
Chia-Pei Tang, Chen-Hung Hsieh and Tu-Liang Lin
Reviewer 2 Report
This manuscript describes the evaluation of several recent image enhancement techniques (Color Transfer with Mean Shift [CTMS], Multi-scale Retinex with Color Restoration [MSRCR], and 15 Gamma and Sigmoid Conversions [GSC]) for enhancing endoscopy images, for improved classification performance. Some general concerns might be addressed before further review:
1. It is strongly considered to clarify the experimental design(s). In particular, two experimental designs are declared (Figure 10 & 11; does not appear fully consistent with Figure 2, which might be removed/reworked). It is not clear as to which of the results in Section 3 relate to these two experimental designs, and what the motivation behind the two designs was. These sections might be reorganized/expanded for clarity.
2. While three new techniques were examined, it might be strongly considered to also compare against some traditional enhancement technique such as CLAHE, to benchmark the contribution of these newer/complex techniques.
3. Minor issues:
(Abstract) Define NBI acronym before first use
(Line 30) The the "resect and discard" and "diagnose and leave" strategies might be described in more detail
(Line 37) "new diagnostic modality has been" -> "new diagnostic modalities have been"
(Line 44) "The NBI remains the most adopted method and widely used at 44 present" -> cite supporting evidence if possible
(Line 64) "The polyp and adenoma detection accuracy are not 64 increased with the NBI system in daily clinical practice" -> explain in greater detail
(Line 70) "not feasible from" -> "not compatible with"?
etc.
Author Response
Dear Editors,
We would like to submit the revised manuscript entitled “Computer-aided image enhanced endoscopy automated system to boost polyp and adenoma detection accuracy”, which we wish to be considered for publication in Diagnostics.
The followings are our responses to the reviewer’s comments.
(1)Reviewer’s Comment: 1. It is strongly considered to clarify the experimental design(s). In particular, two experimental designs are declared (Figure 10 & 11; does not appear fully consistent with Figure 2, which might be removed/reworked). It is not clear as to which of the results in Section 3 relate to these two experimental designs, and what the motivation behind the two designs was. These sections might be reorganized/expanded for clarity.
Authors’ Response: We thank the reviewer for the comment. We have clarified the experimental design as the followings. “In this research, the colorectal polyp identification is performed using two experimental designs. In the first experimental design, as Figure 10, the color transfer is per-formed on the entire colonoscopy images. In the implementation, we found that when the color transfer is performed on the entire high resolution images, it will take some extra time to convert every pixel of the entire image. However, in the actual diagnosis process, it is expected that the system can generate the instant results. Therefore, in order to improve this immediacy problem, this study further proposes a second experimental de-sign, as Figure 11. First the white light endoscopic image is used to identify the polyps, and then only perform color transfer in the identified polyp areas. The entire process takes less time and enhances its immediacy due to a smaller area for color transfer.” We also change the terms so that Figure 10 & 11 is consistent with Figure 2. We also modified the title of section 3.1 and section 3.2 to relate them to the two experimental design. The title of section 3.1 is changed from “YOLOv4 Training Results” to “YOLOv4 Training Results in Experimental Design 1”. The title of section 3.2 is changed from “Classifier Training Results” to “Classifier Training Results in in Experimental Design 2”.
(2)Reviewer’s Comment: While three new techniques were examined, it might be strongly considered to also compare against some traditional enhancement technique such as CLAHE, to benchmark the contribution of these newer/complex techniques.
Authors’ Response: We thank the reviewer for the comment. We have compared the selected methods with two conventional image enhancement methods, Histogram Equalization (HE) and Contrast Limited Adaptive Histogram Equalization (CLAHE). The results of the conventional HE and CLAHE are also included to benchmark with the selected color transfer methods in Table 2, 3, 4, 5, 6, 7.
(3)Reviewer’s Comment: Minor issues:
(Abstract) Define NBI acronym before first use
(Line 30) The the "resect and discard" and "diagnose and leave" strategies might be described in more detail
(Line 37) "new diagnostic modality has been" -> "new diagnostic modalities have been"
(Line 44) "The NBI remains the most adopted method and widely used at 44 present" -> cite supporting evidence if possible
(Line 64) "The polyp and adenoma detection accuracy are not 64 increased with the NBI system in daily clinical practice" -> explain in greater detail
(Line 70) "not feasible from" -> "not compatible with"?
Authors’ Response: We thank the reviewer for the comment. We have modified the content according to the reviewer’s suggestions.
(Abstract) We define the NBI acronym and describe the terms as the following. “Narrowed-band imaging (NBI) is one of the image enhance technique to boost polyp detection and characterization, which uses special filter to enhance the contrast of the mucosa surface and vascular pattern of the polyp.”
(Line 30) We have described the strategies in more detail as the following.
“According to the American Society for Gastrointestinal Endoscopy, the “resect and discard” and “diagnose and leave” strategies propose that the hyperplastic polyp needs not to be removed. Since the hyperplastic polyps are most diminutive and non-malignant, these strategies save the resection time and pathologic analysis cost in a great deal.”
(Line 37) We have modified the sentence according to the suggestion and rewrite it as “As the field of colonoscopy technology thrives, new diagnostic modalities have been introduced to improve polyp detection.”
(Line 44) We have provided the supporting evidence as the following and cited the reference in the manuscript.
Alharbi, O. R., Alballa, N. S., AlRajeh, A. S., Alturki, L. S., Alfuraih, I. M., Jamalaldeen, M. R., & Almadi, M. A. (2019). Use of image-enhanced endoscopy in the characterization of colorectal polyps: Still some ways to go. Saudi Journal of Gastroenterology: Official Journal of the Saudi Gastroenterology Association, 25(2), 89.
(Line 64) This should be “rate” instead of “accuracy”. We have corrected the mistakes in the sentence and rewrite it as “The polyp and adenoma detection rates are not increased with the NBI system in daily clinical practice.”
(Line 70) We have corrected the mistakes in the sentence and rewrite it as “The output connection of the Olympus NBI system from the colonoscopy equip-ment to an external computer is not feasible.”
We thank again for the reviewer’s comment, and we found the comments are very helpful to promote the quality of our paper.
Sincerely,
Chia-Pei Tang, Chen-Hung Hsieh and Tu-Liang Lin
Round 2
Reviewer 2 Report
We thank the authors for addressing our concerns, and particularly for the additional experimental/comparison results. It might however be strongly considered to describe the comparison methods (e.g. HE, CLAHE) more completely in terms of their parameters, to allow for better understanding/replication. Also, "Efficeientnet" -> "Efficientnet".
Author Response
Dear Editors,
We would like to submit the revised manuscript entitled “Computer-aided image enhanced endoscopy automated system to boost polyp and adenoma detection accuracy”, which we wish to be considered for publication in Diagnostics.
The followings are our responses to the reviewer’s comments.
Responses to reviewer 2
(1)Reviewer’s Comment: - We thank the authors for addressing our concerns, and particularly for the additional experimental/comparison results. It might however be strongly considered to describe the comparison methods (e.g. HE, CLAHE) more completely in terms of their parameters, to allow for better understanding/replication. Also, "Efficeientnet" -> "Efficientnet".
Authors’ Response: We thank the reviewer for the helpful comment and spending valuable time to review the paper. We have added the parameter description of HE and CLAHE as the following. “The results of the conventional HE and CLAHE are also included to benchmark with the selected color transfer methods. The images are first transferred from the RGB color space to LAB color space and the HE and CLAHE are performed on the lightness L channel. There are two main operational parameters, tile size and clip limit, in the CLAHE image enhancement process. The tile size is the number of the non-overlapping tile to which the original image is partitioned and is set to 8X8 in the experiments. The clip limit is the threshold that will be used to trim the histograms of the pixel distribution and is set to 2 in the experiments. Also, the “Efficeientnet” has been changed to “Efficientnet”
We thank again for the reviewer’s comment, and we found the comments are very helpful to promote the quality of our paper.
Sincerely,
Chia-Pei Tang, Chen-Hung Hsieh and Tu-Liang Lin